# In Vitro and In Vivo Digestion of Persimmon and Derived Products: A Review

**DOI:** 10.3390/foods10123083

**Published:** 2021-12-11

**Authors:** Cristina M. González, Isabel Hernando, Gemma Moraga

**Affiliations:** Departamento de Tecnología de Alimentos, Universitat Politècnica de València, Camino de Vera s/n, 46022 Valencia, Spain; crima13c@upvnet.upv.es (C.M.G.); gemmoba1@tal.upv.es (G.M.)

**Keywords:** *Diospyros kaki*, bioaccessibility, phenolic compounds, carotenoids, fiber

## Abstract

The link between nutrition and health has focused on the strategy of diet-based programs to deal with various physiological threats, such as cardiovascular disease, oxidative stress, and diabetes. Therefore, the consumption of fruits and vegetables as a safeguard for human health is increasingly important. Among fruits, the intake of persimmon is of great interest because several studies have associated its consumption with health benefits due to its high content of bioactive compounds, fiber, minerals, and vitamins. However, during digestion, some changes take place in persimmon nutritional compounds that condition their subsequent use by the human body. In vitro studies indicate different rates of recovery and bioaccessibility depending on the bioactive compound and the matrix in which they are found. In vivo studies show that the pharmacological application of persimmon or its functional components, such as proanthocyanidins, can help to prevent hyperlipidemia and hyperglycemia. Thus, persimmon and persimmon derived products have the potential to be a fruit recommended for diet therapy. This review aims to compile an updated review of the benefits of persimmon and its derived products, focusing on the in vitro and in vivo digestibility of the main nutrients and bioactive compounds.

## 1. Introduction

Persimmon (*Diospyros kaki L. f*) is a fruit belonging to the Ebenaceae family that originated in China around 450 BC, before spreading to Korea and Japan, where it is considered a traditional crop [1]. Persimmon arrived in Europe at the beginning of the XVII century, with Spain as its main producer. Over the past 19 years, there has been an exponential growth in persimmon production, and around 80% is exported to the European market [2]. According to FAO 2019 data, world persimmon production rose to 4,270,074 tons, with a cultivated area of 992,425 ha. Currently, the largest persimmon producer worldwide is China, followed by Spain, the Republic of Korea, Japan, Brazil, and Azerbaijan. (FAO, 2019). Persimmons are usually classified as astringent and non-astringent varieties. In astringent varieties, persimmons are edible when overripe, but they are astringent at harvest time (when the fruits are still firm). The astringency at the time of harvest is related to the number of soluble tannins accumulated in the fruit, rendering them inedible. They are only edible after a deastringency treatment is applied [3] or when the fruit overripen—as soluble tannins polymerize during the ripening procress [4]. In the non-astringent varieties, the accumulation of soluble tannins ends in the early stages of fruit development, and they are edible at harvest time [5]. Varieties such as *Giombo, Niuxin, Wolha, GongChengYueShi, Yongding*, *Hohrenbo, Ichida-gaki, Gongcheng Yueshi, Tone Wase, Hachiya, Hiratanenashi, Jiangsu, Atago, Aodanshi, Triumph, Rojo Brillante, Mopan,* and *Sangju* are astringent, whereas *Fuyu*, *Hana-Fuyu*, *Cal-Fuyu*, *O’Gosho*, *Hana-Gosho*, *Eshi*, *Jiro*, and *Kaki Tipo* are non-astringent [1,5,6]. *Tone Wase, Hachiya,* and *Saijo* varieties are widely cultivated in Japan; *Fuyu*, *Hana-Fuju* or *Jiro*, *GongChengYueShi, Yongding*, *Hohrenbo, Ichida-gaki,* and *Mopan* are cultivated in China; *Kaki Tipo* is typical in Italy; and *Rojo Brillante,* along with *Triumph,* are grown in Spain. The exponential growth of persimmon in Spain is mainly because of the introduction of the astringent variety *Rojo Brillante*, because of its excellent quality and adaptation to Spain’s climate conditions [7].

The cultivation of persimmon has a limited shelf life compared to other fruits. It is a seasonal fruit and is perishable and difficult to store and transport; therefore, many persimmons are discarded [8]. Hence, it is necessary to search for alternatives for the use and valorization of the discarded persimmon cultivars. Dehydrated persimmons are commonly commercialized in countries such as China, Korea, and Japan [9]. Products derived from persimmon such as persimmon flour have also been obtained [10] and used in pork liver pâté [11] or pasta formulation [12]. Ice cream prepared with persimmon puree [13], dairy products accompanied by persimmon [14], persimmon syrup [15], vinegar [16], persimmon wine [17], and persimmon snacks obtained by different drying treatments [18,19] have also been prepared in different studies.

Both persimmon and the derived products have a high content of fiber, minerals, vitamins, and bioactive compounds, which gives different beneficial effects against diseases such as oxidative stress, hypertension, diabetes mellitus, and atherosclerosis [20]. However, changes take place during digestion in the bioactive and nutritional compounds affecting the subsequent use of these substances by the human body. Determining food digestibility is conducted using concepts such as bioavailability, bioaccessibility, and bioactivity of food components. Bioavailability, the ingested fraction of a biocomponent that reaches the systemic circulation to be distributed to organs and tissues, is determined using in vivo studies. Bioaccessibility, the ingested fraction of a biocomponent that becomes accessible for absorption through the epithelial layer of the gastrointestinal tract, is determined using in vitro studies. The bioactivity represents the ability of a compound to manifest a biological effect [21].

This review aims to compile the current information on in vitro and in vivo digestion studies of persimmon and its derived products, focusing on the main bioactive and nutritional compounds.

## 2. In Vitro Digestion Studies of Persimmon and Derived Products

In vitro digestion is widely used to study the gastrointestinal behavior of foods. It reproduces the human physiological gastrointestinal process in the laboratory in a controlled and reproducible way [22,23]. Most literature related to the in vitro digestion of persimmon and its derived products focus on the digestibility of phenolic compounds and, to less extent, on the digestibility of carotenoids (Table 1).

### 2.1. Phenolic Compounds Digestibility

Phenolic compounds are secondary metabolites with an important antioxidant activity that can be classified as extractable polyphenols (EP) and non-extractable polyphenols (NEP). EP is easily extracted with solvents, whereas NEP need acidic or basis hydrolysis to be broken, so they can be extracted and quantified in vitro.

#### 2.1.1. Persimmon Fruit

Martínez-Las Heras et al. [24] analyzed the content of EP and the antioxidant capacity of the fruit, leaves, and fiber extracted from *Rojo Brillante* persimmon during its in vitro digestion. The results showed that the oral phase and the presence of α-amylase were the factors that most affected the reduction in the recovery index of the EP, especially in the leaves of persimmon. An increase in the recovery index of EP was observed in persimmon fruit and fibers during the small intestine phase. This could be because of the extended period of this phase (>2 h) and the effect of intestinal enzymes and bile salts, which could facilitate the release of polyphenols from the persimmon matrix. However, antioxidant capacity during digestion resulted in total losses at the end of digestion in leaves, persimmon fruit, and fibers. The bioaccessibility of the EP in the persimmon fiber was higher than in the fruit and persimmon leaves. Moreover, the bioaccessibility of the total antioxidant capacity was lower than those of EP and never exceeded 40%. They concluded that the EP and total antioxidant capacity of the aqueous extract of persimmon leaf were more sensitive to the gastrointestinal environment than those derived from persimmon fruits or fibers. Although the bioaccessibility of the total antioxidant compounds in the persimmon fruit and the fiber was greater than in the aqueous extract of persimmon leaves, an infusion with persimmon leaf (1.5 g in 110 mL of water) and persimmon fruit (200 g) would provide similar bioaccessible antioxidants at the end of digestion.

Zhu et al. [25] analyzed the inhibitory effect of tannins extracted from persimmon *Niuxin* on pancreatic lipase. This critical enzyme is associated with hyperlipidemia and obesity. The results showed the tannins extracted from persimmon had a high affinity for pancreatic lipase and inhibited the activity of this enzyme; the interaction was spontaneous through non-covalent bonds. Therefore, the binding and inhibition capacity of persimmon tannins on lipid digestive enzymes may have effectiveness for the treatment and prevention of obesity.

Li et al. [26] showed the effect of the tannins extracted from the fresh *GongChengYueShi* persimmon on the digestibility of corn starch and on the activity of α-amylase and α-glucosidase; two of the main digestive enzymes involved in the hydrolysis of the starch. The digestibility of starch decreased with the addition of tannins from persimmon; the higher the concentration of added persimmon tannins, the greater the inhibition of starch digestibility. Moreover, the results showed persimmon tannins interacted with the starch, interacting with amylose more than amylopectin. However, tannins exerted a great inhibitory effect on α-amylase and α-glucosidase. Generally, data suggest that tannins in persimmon may help reduce postprandial hyperglycemia, regulating glucose levels in the human body.

Lee et al. [27] evaluated the polymers and oligomers of proanthocyanidins from persimmon peel against diabetes. The oligomeric proanthocyanidins exerted a higher interaction with α-glucosidase and the polymeric proanthocyanidins had greater interaction with α-amylase. This suggests that the inhibition of both enzymes depends on the degree of polymerization of the phenols. Therefore, persimmon peel could have an antidiabetic action.

In a study conducted by Zhou et al. [8], the total content of EP and NEP present in *Mopan* persimmon was evaluated. The action against α-glucosidase of EP and NEP and their antioxidant capacity were also compared. In addition, the release of EP from NEP was studied throughout in vitro digestion, considering the oral, gastric, and small intestinal phases. The results showed the NEP content in fresh persimmon was higher than EP. The EP had a greater inhibition capacity of α-glucosidase than the NEP. This could be because EP comprised low molecular weight molecules, which can be more easily attached and have a greater ability to interact with the enzyme. Both had a higher α-glucosidase inhibition capacity than acarbose—an oligosaccharide used as a drug reducing the speed of carbohydrate digestion. After in vitro digestion of NEP, the polyphenols content and the antioxidant capacity were lower in the oral phase and significantly increased in the gastric and intestinal phases. The acidic conditions of the stomach environment can enhance the release of NEP. Therefore, NEP was released after simulated gastrointestinal digestion, and the gastric phase played a key role in their release. Thus, NEP was the most effective antioxidant in persimmon fruit after digestion. Nevertheless, Zhou et al. [8] suggested further investigations to explain the composition and structure of EP and NEP to clarify their differences in biological activities.

#### 2.1.2. Persimmon Derived Products

Several authors have studied the in vitro digestibility of polyphenols in products such as persimmon flour, spaghetti, and pork liver pâté enriched with persimmon flour, persimmon peels, and dehydrated persimmon. Lucas-González et al. [28] evaluated the recovery index, bioaccessibility of EP, and total soluble flavonoids during the in vitro digestion of persimmon flour derived from the *Rojo Brillante* and *Triumph* varieties. The recovery index of the EP in both varieties was similar, except in the oral phase, where *Triumph* showed lower values, and the recovery index of total soluble flavonoids was higher in the *Rojo Brillante* variety. These could be related to the different compositions of the persimmon varieties (total dietary fiber is higher in the *Triumph* variety) and the interaction of phenolic compounds with α-amylase. The highest recovery index was obtained after the gastric phase and the lowest after the small intestine phase in both varieties. This could be explained by the different pH in the digestion phases. At the acidic pH present in the stomach, phenols are often found in a very stable chemical form—the Flavylium cation. Therefore, the bioaccessibility in the stomach could be higher. However, this bioaccessibility drastically decreases in the intestine because of interactions with dietary compounds such as fiber, chemical reactions of oxidation, and polymerization, or molecular changes because of enzyme action. Regarding the bioaccessibility of EP, the results indicated both flours presented similar values. *Triumph* presented greater bioaccessibility of soluble flavonoids than the *Rojo Brillante* variety, probably because of the different flavonoid profiles of the persimmon flours. Therefore, the different fiber content, the interaction of α-amylase with polyphenols, the pH during the gastric and intestinal phase, and the total content of polyphenols and flavonoids in the persimmon samples are the main factors affecting the behavior of these bioactive compounds during digestion [28].

In another study, Lucas-González et al. [12] formulated spaghetti with 3% and 6% of the persimmon flours obtained from the *Rojo Brillante* and *Triumph* varieties. The profile of EP and NEP and their bioaccessibility and antioxidant capacity after simulated in vitro digestion were determined. Spaghetti enriched with persimmon flours modified the polyphenolic profile with the appearance of two new compounds, gallic acid and p-coumaric-o-hexoside, increasing the antioxidant capacity. After in vitro digestion, numerous polyphenols remained bound to the cell wall or to indigestible polysaccharides. The EP bioaccessibility determined in the small intestine phase was poor and did not improve with the addition of persimmon flours; many EP could become part of the NEP. The authors concluded that although most NEP did not release from the food matrix during gastrointestinal digestion, they may still have a health-promoting effect as they could be available in the colon [12].

Furthermore, Lucas-González et al. [29] enriched pork liver pâté with 3% and 6% of *Rojo Brillante* persimmon flour. In both samples of enriched pâté, 2 EP and 21 NEP were detected, provided by the persimmon flour. After in vitro digestion of the pâté samples, EP and NEP were evaluated. The in vitro digestion consisted of the oral, gastric, and two small intestine phases; one with pancreatin high lipase activity (C1) and the other with pancreatine low lipase activity (C2). More NEP than EP was detected in all digestion stages. In addition, it was observed that the intestinal phase C1 was more suitable to recover NEP after digestion than the intestinal phase C2. This could be associated with a greater release of fatty acids in the digestive environment, which could have a protective effect on polyphenols by interacting with them. However, the observed polyphenols were NEP, which were not released. Therefore, they probably reached the colon intact, and some could be metabolized by the intestinal microbiome. Lucas-González et al. [29] concluded that high-fat foods such as pâté are excellent vehicles for preserving NEP, which could reach the colon intact and be metabolized by the intestinal microbiome. However, more studies are needed on lipid digestibility, colonic fermentation, and polyphenol transformations to achieve the complete health implications of fortifying meat products with persimmon flours.

Liu et al. [30] selected optimal deastringency methods and evaluated the bioaccessibility of polyphenols in *Yongding* persimmon peels treated with 12 combinations of CO_2_ and ethanol. EP and NEP content after the deastringency treatments as well as the antioxidant capacity and bioaccessibility of the EP after in vitro digestion, were determined. The results indicated that the EP content decreased, and the NEP content increased with increasing ethanol and CO_2_ concentration in the non-digested samples. After in vitro digestion, there was also a significant decrease in the EP and antioxidant capacity of persimmon peels. This could be related to a higher NEP formation after the digestion process. The ethanol (30%) and CO_2_ vapor (70%) methods were the most effective, with the highest bioaccessibility of EP. Therefore, it could be considered the best deastringency method.

Recent studies investigated the recovery index of EP, soluble flavonoids, and antioxidant capacity of persimmon affected using drying methods—ultrasound-assisted vacuum drying (USV), freeze-drying (FD), infrared drying (ID), and hot air drying (HAD) [31]. The results showed that the USV, ID, and HAD led to a significant increase in the bioaccessibility of EP, soluble flavonoids, and antioxidant capacity compared to fresh persimmon. The samples obtained by FD did not show significant differences in the recovery index with respect to fresh persimmon. The recovery index increment was produced because of the heat treatment during the drying process. The heat facilitated the release of bioactive compounds from the food matrix. This would also explain the higher bioaccessibility obtained for ID and HAD because they are the drying processes with the highest thermal load. Therefore, the best dehydration processes to obtain higher bioaccessibility of EP, soluble flavonoids, and antioxidant capacity are HAD and ID.

Bas-Bellver et al. [32] analyzed the effect of the gastric, small intestine phase, and colonic fermentation on the phenolic compounds and the antioxidant capacity of *Rojo Brillante* persimmon powders obtained using the HAD and FD methods. In other studies, EP and antioxidant capacity increased after the gastric phase and decreased after the intestinal phase. As they stated, most of the solubilized polyphenolic compounds remained in the precipitate. The bioaccessibility of the EP of both samples obtained by both drying treatments showed no differences between them, whereas the bioaccessibility of the antioxidant capacity increased in the samples treated by freeze-drying. After the colonic fermentation of the predigested samples, a growth of beneficial bacteria such as *Bifidobacterium* and *Faecalibacterium prausnitzii* was observed. Likewise, positive correlations were detected between polyphenols and *Actinobacteria, Akkermansia,* and *Ruminococcaceae*; bacteria genera related to beneficial effects on the immune system and health status. Lactic acid bacteria, *Streptococcus,* and *Lactobacillus* also showed higher abundance after fermenting the digested persimmon samples. In addition, butyrate-producing bacteria such as *Faecalibacterium* and *Ruminococcaeae* also showed higher abundance after fermentations. Butyrate is a short-chain fatty acid produced by intestinal bacteria because of the fermentation of indigestible polysaccharides. This metabolite is a critical mediator of the colonic inflammatory response and a contributor to the immune system. Thus, persimmon powders could be used in food formulation to improve the content of bioactive compounds and could influence human health.

Matsumura et al. [33] investigated the in vitro antioxidant potential of NEP from *Hohrenbo* dehydrated persimmon using HAD. They performed the in vitro digestion divided into four phases: oral, gastric, small intestine, and large intestine and determined the antioxidant capacity in each phase. The antioxidant capacity in the oral phase was low but increased in the gastric and small intestine phases. However, the highest values of antioxidant capacity were obtained in the large intestine phase. In the large intestine phase, the intestinal microflora produced the fermentative decomposition of the non-extractable fraction of the dried persimmon, enhancing its antioxidant capacity. Moreover, the authors concluded that more studies are required to confirm the health benefits of NEP and to distinguish between the dietary functions of EP and NEP.

Hamauzu and Suwannachot [34] analyzed the EP and NEP fractions in *Ichida-gaki* persimmon samples dehydrated using natural drying. After simulating the gastric and small intestine phase in vitro, the NEP fraction presented a strong bile acid-binding capacity. Therefore, dehydrated persimmon with a large amount of NEP could act as a cholesterol-lowering agent.

### 2.2. Carotenoids Digestibility

To learn about the beneficial properties of persimmon carotenoids for human health, derived from their antioxidant capacity and provitamin A function, several studies have evaluated the bioaccessibility and stability of persimmon carotenoids during digestion. 

#### 2.2.1. Persimmon Fruit

Estévez-Santiago et al. [35] evaluated the bioaccessibility of provitamin A carotenoids from different fruits, including persimmon. The carotenoids evaluated were β-cryptoxanthin, β-carotene, and α-carotene in their trans/cis forms using HPLC quantification. The carotenoids bioaccessibility in persimmon was low, where β-carotene and α-carotene had the highest percentage of bioaccessibility. Estévez-Santiago et al. [35] explain that the effect of the food matrix affects the bioaccessibility of carotenoids, and bioaccessibility increases with the presence of a fat source. Fruit is rarely consumed with a fat source, as this mixture is not a common food offering. However, fruits are usually eaten as desserts after meals containing fat. This practice can have a positive effect on the bioaccessibility of carotenoids.

According to Cano et al. [36], the bioaccessibility of carotenoids involves two processes: (i) release of carotenoids from the food matrix and (ii) subsequent micellization. This is limited by many factors, such as the presence of lipids, processing (milling, mechanical grinding), or the type of food matrix. Only the carotenoids present in the micellar phase are considered bioaccessible. Cano et al. [36] evaluated the stability and bioaccessibility of carotenoids and carotenoid esters in the *Rojo Brillante* persimmon in fresh fruit and after high hydrostatic pressure and pasteurization treatments. The number of carotenoids before and after digestion was quantified using the HPLC technique, and their recovery and bioaccessibility indexes were calculated after the oral, gastric, and small intestine phases. The results in fresh persimmon showed low bioaccessibility and no micellization of carotenoids; traces were found in the micellar phase of the small intestine phase. This was related to the fiber content of persimmon that traps bioactive compounds and reduces micellization and bioaccessibility. Pressurized and thermally treated samples increased the overall carotenoid bioaccessibility to 54% and 25%, respectively. This increase in bioaccessibility could be because of structural modification (pressurized samples) or degradation plant of polysaccharides (pasteurized samples), such as pectin—present in persimmon tissue—releasing the carotenoids and favor the subsequent micellization.

#### 2.2.2. Persimmon Derived Products

García-Cayuela et al. [37] assessed the in vitro carotenoids recovery index and bioaccessibility in *Rojo Brillante* persimmon-based dairy products. Dairy products were formulated with whole milk (3.6% fat) or skimmed milk (0.25% fat) and with whole freeze-dried persimmon, pulp, or peel. On average, the total carotenoid recovery was approximately between 25–39%. This means that the total carotenoid content decreased between 66–75% after in vitro digestion in all the formulated samples. The carotenoids bioaccessibility was significantly higher in dairy products formulated with whole milk. Within the whole milk formulations, the highest amount of bioaccessible carotenoids was provided by dairy products, including peel, followed by those including whole persimmon and those with the pulp. Furthermore, these formulations significantly improved the bioaccessibility of provitamin A total carotenoids (β-cryptoxanthin, α-carotene, β-carotene, and lycopene). García-Cayuela et al. [37] suggested, as did Estévez-Santiago et al. [35], that a higher fat content in the product exerts a significant improvement in carotenoid bioaccessibility. The highest amounts of bioaccessible carotenoids were found in whole milk + whole persimmon and whole milk + peel. Therefore, these formulations would be the most suitable for developing functional foods for people with low vitamin A consumption.

Bas-Bellver et al. [32] determined the changes in persimmon carotenoids after FD and HAD treatments and during in vitro digestion. The total carotenoid content was slightly higher in the FD powders than in the HAD because of the absence of oxygen and the low temperature during the FD treatment. α-Cryptoxanthin was the most abundant carotenoid in both persimmon powders. Nevertheless, the degradation of the carotenoids analyzed was evidenced during in vitro digestion in both drying treatments. This degradation during digestion depended on the content and characteristics of fiber and lipids present in the food matrix. After colonic fermentation, the number of certain beneficial bacteria genera were slightly greater with the presence of the persimmon powder samples. There was a positive correlation with beneficial bacteria genera and a negative correlation with harmful bacteria, thus indicating that the presence of antioxidant compounds (polyphenols and carotenoids) is associated with a high and low abundance of these genera. Therefore, persimmon waste powders could be included in the food formulation to improve the content of carotenoids and could have a positive effect on human health.

## 3. In Vivo Digestion Studies on Persimmon and Derived Products

In vitro studies are often complemented by in vivo studies since they are more representative considering the complexity of organisms. The studies found regarding in vivo digestion on persimmon and derived products are focused on evaluating the beneficial effects on lipid metabolism, the regulation of glucose levels, and carcinogenic and anti-inflammatory effects.

Table 2 summarizes the studies related to the in vivo digestion of persimmon and their derived products.

### 3.1. Effect on Lipid Metabolism

To determine the effect of persimmon derived products and phenolic compounds, such as tannins—on lipid metabolism—several markers have been measured. Based on the literature, the most common markers are cholesterol (TC), high-density lipoprotein (HDL), low-density lipoprotein (LDL), and triglycerides (TG).

Gorinstein et al. [38,39,40] observed a hypocholesterolemic and antioxidant effect in rats fed with dried persimmon (peel, pulp, and different varieties such as *Fuyu* and *Jiro*). The diets supplemented with persimmon presented a lower increase in TC, LDL, HDL, and TG markers compared to the control diet. Persimmon peel and pulp could help prepare new foods in industrial processes lowering hyperlipidemia parameters in the consumers. Kim et al. [41] studied the anti-obesity effect of an extract made with persimmon fruit (*Sangju*) and satsuma mandarin peel incorporated into a high-fat diet in mice. They observed that the extract inhibited triglyceride absorption by inhibiting pancreatic lipase, and a preventive effect on the visceral fat accumulation was also observed.

Matsumoto et al. [42,43,44] and Matsumoto and Takeawa [45] studied the effect of dried mature persimmon (MP) and dried young persimmon (YP) (non-astringent *Fuyu* and astringent *Hachiya,* and *Hiratanenashi*) on lipid metabolism. The main conclusions from these researchers were that none of the MP fed mice improved lipid metabolism; however, YP exerted beneficial effects such as preventing hepatic steatosis and reducing plasma cholesterol. Besides, the *Hachiya* YP-fed groups showed lower levels of TC, TG, LDL, and HDL. Matsumoto et al. [44] observed that tannins from *Hachiya* persimmon had a high affinity for bile acids and significantly promoted their fecal excretion. Therefore, tannins would be beneficial compounds in the prevention and improvement of metabolic syndrome. Matsumoto and Takeawa [45] confirmed that astringent persimmon with a high content of soluble tannins improves plasma cholesterol conditions. *Hachiya,* the cultivar that obtained the highest bile acid-binding capacity and the highest content of soluble tannins, can be considered the best to maintain good health. These studies demonstrate that the beneficial effects of persimmon on lipid metabolism are related to bile acid-binding capacity and soluble tannin content. Zou et al. [46] observed that tannins were mainly responsible for the antihyperlipidemic effect of persimmon. Besides, diets supplemented with tannins decreased the levels of the enzyme fatty acid synthase (FAS) responsible for catalyzing fatty acids and stimulating the genes responsible for lipogenesis and suppressed tumor necrosis factor (TNFα) and C-reactive protein (CRP) responsible for regulating inflammation.

Zhu et al. [47] evaluated the effect of *Niuxin* persimmon tannins on the intestinal microbiota. The diet with tannins altered the composition of the intestinal microbiota by increasing the bacteroidetes/proteobacteria ratio (bacteria inversely related to obesity). Therefore, the hypolipidemic effect of polyphenols could be attributed partially to the significant improvement of the microbial composition of the intestine contributing to health.

Some persimmon derivatives such as white wine, persimmon vinegar, and peel powder have also prevented an increase of TG and TC levels in rabbits, hamsters, and rats, demonstrating an antiatherogenic effect and reducing the risk of cardiovascular diseases [16,48,49].

Regarding the fiber present in persimmon, Gato et al. [50] showed plasma cholesterol levels decreased significantly in humans fed a tannin-rich fiber diet. Lee et al. [51] showed that food intake and body-weight gain were reduced with a diet supplemented with persimmon leaf (*Sangju*). They related these results to the fiber content acting as a satiating agent. A reduction in TC, HDL, and hepatic TG was also observed, and feces had a higher TG content in the diet supplemented with persimmon leaf.

Regarding the antiadipogenic effect, Shin, Shon, Kim, and Lee [52] determined whether extractable and non-extractable tannin extracts suppressed adipogenesis or the conversion of preadipocytes into adipocytes. 3T3-L1 cells were treated with extractable and non-extractable tannins from five types of persimmons. Treatment with extractable and non-extractable tannins for 7 days resulted in a significant inhibition of adipogenesis. Therefore, tannin extracts could inhibit or alter the expression of specific genes involved in adipogenesis, although it is not clear how extractable and non-extractable tannin extracts regulate the adipogenesis process.

### 3.2. Antidiabetic Effects

Diets supplemented with persimmon peel [53], persimmon tannins [26,54], and persimmon extract [55] improved characteristic symptoms of type II diabetes such as insulin resistance, hyperphagia, and the decrease of glucose blood levels delaying glucose into the bloodstream. The antihyperglycemic properties observed could be because of the combined effect of dietary fiber and antioxidants such as polyphenols and carotenoids present in persimmon. Shin et al. [55] also observed the inhibition capacity of the digestive enzymes α-amylase and α-glucosidase, as seen in Section 2.1.1. Besides, persimmon extract also enhanced the cholinergic system damaged by oxidative stress and the endogenous antioxidant system in brain and liver tissues induced by the high-fat diet. Therefore, these results indicated that persimmon, persimmon tannins, and persimmon extracts could be a good dietary supplement for the synthesis of antidiabetic drugs; and could have the potential as a natural functional food material to improve cognitive functions.

### 3.3. Anti-Carcinogenic and Anti-Inflammatory Effects

The possible beneficial effect of persimmon on carcinogenic cells and anti-inflammatory effects in different diseases has also been studied. Yumiko et al. [56] showed phenolic compounds from persimmon such as catechin, epicatechin, epigallocatechin, and epigallocatechin gallate inhibited the growth of lymphoid cells. Furthermore, DNA fragmentation of the treated cells was observed, suggesting these compounds induce the death of carcinogenic cells.

Kawase et al. [57] evaluated cytotoxic activity, activity against the human immunodeficiency virus (HIV), and *Helicobacter pylori* of persimmon peel (*Aodanshi*). The results indicate the existence of useful therapeutic bioactive compounds and the therapeutic value of persimmon peel extract. However, these fractions should be further purified to identify the main compounds related to a possible antitumor agent.

Direito et al. [58] recently determined the anti-inflammatory effects of persimmon extract on rheumatoid arthritis disease through in vivo digestion in mice. A reduction in both edema and alterations caused by arthritis was observed after the consumption of the persimmon extract. Therefore, the administration of persimmon extracts attenuated inflammation and tissue damage. This could be because of the powerful antioxidant characteristics that persimmon presents. The consumption of persimmon extracts can also be a useful pharmacological tool in the management of chronic arthritic conditions associated with active inflammation.

## 4. Conclusions

In recent years, persimmon production has increased, and the beneficial effects of its nutrients and bioactive compounds have gained a lot of interest; phenolic compounds and carotenoids have shown the greatest promise related to human health. The digestibility of these compounds throughout the gastrointestinal tract (oral, gastric, and intestinal phases) is mainly affected by the food matrix, enzymes, pH, and digestive fluids. In vitro studies show that the gastric phase plays an important role by increasing the release of EP mainly due to the acidic conditions of the stomach environment. However, the small intestine phase produces a reduction of EP because of interactions with dietary fiber, chemical reactions such as polymerization, or molecular changes by the action of bile salts and intestinal enzymes. Drying treatments with high temperatures increased the recovery index and the antioxidant properties of the EP, favoring their release from the persimmon matrix. NEP with high antioxidant potential can reach the colonic phase intact; thus, they can interact with fiber and perform their function on the intestinal microbiota. The intake of persimmon with foods rich in fat and the introduction of derivatives of persimmon in matrices with a high percentage of lipids are key factors for increasing carotenoids bioaccessibility. 

In vivo studies show that the bioactive compounds of persimmon and, specifically, tannins have beneficial effects because of their high antioxidant and inhibitory capacity against the enzymes responsible for the transport and absorption of glucose and fat during digestion. These effects produce benefits in human health such as helping to reduce blood cholesterol levels, induce tumor cell death, help prevent cardiovascular diseases, regulate diabetes, and adipogenesis.

## Figures and Tables

**Table 1 foods-10-03083-t001:** Studies related to the in vitro digestion of persimmon and derived products.

Food Matrix	Variety	In Vitro Method	Analytical Method	Outcomes	References
Phenolic compounds					
Fiber, fresh fruit, and persimmon leaf	*Rojo Brillante*	Oral, gastric, and small intestine phases	EP and soluble flavonoids. Antioxidant capacity	The oral phase and α-amylase decreased the recovery index of EP. The intestinal phase increased the recovery index. The final bioaccessibility of the phenolic compounds was improved.	[24]
Persimmon tannins	*Niuxin*	Pancreatic lipase activity inhibition	Spectrophotometry analysis	Inhibitory effect of tannins on pancreatic lipase. Hypolipidemic effect.	[25]
Persimmon tannins	*GongChengYueShi*	In vitro starch digestibility	α-Amylase, α-Glucosidase activity assay interaction of tannin with starch.	Tannins help prevent postprandial hyperglycemia.	[26]
Polymers and oligomers of proanthocyanidins from persimmon peel	*-*	-	α-Amylase, α-Glucosidase activity assay	Oligomers have inhibitory force on α-glucosidase and polymers have inhibitory force on α-amylase. Both have antidiabetic action.	[27]
Persimmon fruit	*Mopan*	Oral, gastric, and small intestine phases	Total phenol content (EP and NEP), antioxidant capacity, α-Glucosidase inhibition activity.	EP and NEP inhibit α-glucosidase. NEP were released and the gastric phase played a key role in their release.	[8]
Persimmon flours	*Rojo Brillante* and *Triumph*	Oral, gastric, and small intestine phases	EP and soluble flavonoids.	The fiber content, α-amylase interactions, pH differences, and polyphenols in the sample are the key factors that affect bioactive compounds during digestion.	[28]
Spaghetti with 3% and 6% of persimmon flours	*Rojo Brillante* and *Triumph*	Oral, gastric, and small intestine phases	Total phenol content (EP and NEP) antioxidant capacity	Both persimmon flours added in 3% could develop spaghetti with higher polyphenol content. Bound polyphenols continue to the colon, being used by the intestinal microbiota.	[12]
Pork liver pâté with 3% and 6% of persimmon flours	*Rojo Brillante*	Oral, gastric, and small intestine phases	Total phenol content (EP and NEP)	NEP reach the colon intact and could be metabolized by the intestinal microbiota.	[29]
Persimmon peels	*Yongding*	Oral, gastric, and small intestine phases	Total phenol content (EP and NEP) antioxidant capacity	EP decrease and NEP increase. 30% ethanol and 70% CO_2_ improved the bioaccessibility of total polyphenols and antioxidant properties.	[30]
Dehydrated persimmon	Local market (Turkey)	Oral, gastric, and small intestine phases	EP and soluble flavonoids. antioxidant capacity	The higher temperature of drying, the higher increase of EP, soluble flavonoids, and antioxidant capacity bioaccessibility.	[31]
Persimmon powders obtained using hot air drying (HAD) and freeze-drying (FD) treatments	*Rojo Brillante*	Oral, gastric, and small intestine phases In vitro colonic fermentation	Total phenol content (EP and NEP) antioxidant capacity. characterization of microbiota.	Polyphenols and antioxidant capacity increased after the gastric phase and decreased after the intestinal phase. Positive correlations between polyphenols and bacteria genera after colonic fermentation.	[32]
Dehydrated persimmon	*Hohrenbo*	Oral, gastric, small intestine and large intestine phases	NEP determination Antioxidant capacity	NEP fermentative decomposition in the large intestine. The antioxidant properties increase.	[33]
Dehydrated persimmon	*Ichida-gaki*	Gastric and small intestine phases	Total phenol content (EP and NEP). In vitro acid-binding capacity.	Strong bile acid-binding activity of NEP. Dried persimmon as cholesterol-lowering agents.	[34]
Carotenoids					
Persimmon fruit		Oral, gastric, and small intestine phases	HPLC analysis of carotenoids.	The bioaccessibility of carotenoids increases with the presence of a fat source.	[35]
Persimmon fruit	*Rojo Brillante*	Oral, gastric, and small intestine phases	HPLC analysis of carotenoids.	The pressurization and pasteurization processes increase the bioaccessibility of persimmon carotenoids.	[36]
Persimmon-based dairy products	*Rojo Brillante*	Oral, gastric, and small intestine phases	HPLC analysis of carotenoids	Higher bioaccessibility of carotenoid in dairy products formulated with whole milk.	[37]
Persimmon powders obtained by HAD and FD treatments	*Rojo Brillante*	Oral, gastric, and small intestine phases In vitro colonic fermentation	HPLC analysis of carotenoids. Characterization of microbiota.	The degradation of the carotenoids was evidenced in HAD and FD treatments. Greater positive bacteria genera in the colon.	[32]

EP (Extractable Polyphenols), NEP (Non-extractable Polyphenols), HAD (Hot air-drying), FD (Freeze-drying).

**Table 2 foods-10-03083-t002:** Studies related to the in vivo digestion of persimmon and derived products.

Food Matrix	Variety	In Vivo Method	Health Benefits	References
Effect on lipid metabolism				
Dehydrated persimmon		Rats fed with 7% persimmon	Lower increase in TC, HDL, LDL, and TG levels. Hypolipidemic effect and antioxidant properties.	[38]
Dehydrated persimmon		Rats fed with 7% persimmon peel and pulp	Persimmon peel has a greater hypocholesterolemic and antioxidant effect than persimmon pulp.	[39]
Persimmon freeze-dried	*Fuyu* and *Jiro*	Rats supplemented with 5% freeze-dried persimmon	*Fuyu* and *Jiro* varieties can be considered as hypocholesterolemic as they helped lower blood TG levels.	[40]
Persimmon and satsuma mandarin peel extract	*Sangju*	Mice fed 50 and 200 mg/kg/day fruit extract	Extract with persimmon fruit and satsuma mandarin peel could attenuate some of the physiological changes that occur in obesity and be an anti-obesity agent.	[41]
Young persimmon fruits	*Hachiya*	Mice fed 2% and 5% persimmon	Lower increase of TC, TG, HDL, and LDL levels. Young persimmon contributes to hypolipidemic effect.	[42]
Persimmon tannins	*Hachiya*	Mice fed 1% (*w*/*w*) of tannins	Tannins can help in the prevention and improvement of metabolic syndrome.	[43]
Mature and young persimmon fruit	*Fuyu-kaki* and *Hachiya-kaki*	Mice fed persimmon *ad libitum*	The young persimmon fruits exert beneficial effects on hepatic steatosis, plasma cholesterol, and dyslipidemia.	[44]
Three young persimmon fruits	*Fuyu Hiratanenashi* and *Hachiya*	Mice supplemented with 2% of each persimmon	*Hachiya* was considered the best of the three cultivars to maintain good health.	[45]
Tannins and freeze-dried whole persimmon	*Gongcheng Yueshi*	Rats fed 0.5% tannins extracted from persimmon and with 4.2% freeze-dried persimmon	Tannins were mainly responsible for the antihyperlipidemic effect of persimmon.	[46]
Persimmon tannin	*Niuxin*	Mice fed different dose of tannins (0, 50, 100, and 200 mg/kg weight)	Tannins altered the composition of the intestinal microbiota by increasing the bacteria inversely related to obesity.	[47]
Alcohol-free persimmon white wine		Hamsters supplemented with 7 mL/kg/day wine	Persimmon wine produce antiatherogenic and antioxidant effects.	[48]
Persimmon vinegar		Mice with chronic alcoholism supplemented with 1 mL and 2 mL/ kg of body weight vinegar	Persimmon vinegar prevents alcohol-induced metabolic disorders.	[16]
Persimmon peel powder		Rabbits supplemented with 0, 10, and 20% persimmon	The persimmon peel powder reduces the levels of TC, TG, and LDL cholesterol.	[49]
Persimmon fiber rich in tannins	*Hiratanenashi*	Humans fed cookies supplemented with persimmon fiber (0, 3, 5 g of fiber)	The tannin-rich fiber of persimmon is a useful dietary component to treat hypercholesterolemia.	[50]
Persimmon peels powders	*Sangju*	Rats supplemented with 5% persimmon	Supplementation of powdered persimmon leaf suppress body-weight gain, reduced plasma and liver lipid concentrations, and increase the fecal lipids.	[51]
Soluble and insoluble tannins extracted from persimmon		Key cells inhibition in adipogenesis	Tannin extracts could inhibit or alter the expression of specific genes involved in the adipogenesis.	[52]
Antidiabetic effects				
Persimmon peels	Local persimmon production	Diabetic rats supplemented with 5% and 10% persimmon	Useful dietary supplement for the synthesis of antidiabetic drugs.	[53]
Persimmon tannins	*Atago*	Rats supplemented by 100, 200, and 300 mg/kg body weight tannins	Reduction of postprandial hyperglycemia.	[54]
Persimmon tannins	*GongChengYueShi*	Rats administrated with 0, 25, 50, and 75 mg/kg body weight tannins	Persimmon tannins can alleviate postprandial hyperglycemia.	[26]
Persimmon extract		Mice administrated with 50 and 100 mg/kg persimmon	Persimmon extract has the potential to be a natural functional food material for improved cognitive function.	[55]
Anti-carcinogenic and anti-inflammatory effects				
Persimmon extract		Cell culture. Human Lymphoid leukemia DNA fragmentation assay	Polyphenols induce the death of carcinogenic cells.	[56]
Persimmon peel	*Aodanshi*	Anti-H. pylori activity. Assay anti-HIV activity Cytotoxic activity	Persimmon peel is a possible antitumor agent.	[57]
Persimmon extract		Rats treated with 15 mg crude extract per kg per day	Reduction of arthritis symptoms was observed with the administration of the persimmon extract.	[58]

TC (Total Cholesterol), HDL (High Density Lipoprotein), LDL (Low Density Lipoprotein), TG (Triglycerides).

## Data Availability

Not applicable.

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
