# Peer review of "In Vitro and In Vivo Digestion of Persimmon and Derived Products: A Review"

_foods, 2021, doi:10.3390/foods10123083_

Round 1

Reviewer 1 Report

The parts (points) which was hard to understand in the first version is considered to be improved in revised article, So, revised article is expected to become help for people and researches who wanted expand their knowledge about persimmon and its products.

Author Response

Thank you to the reviewer's kind comments. We are glad to have solved the initial misunderstanding.

Reviewer 2 Report

The manuscript entitled "In vitro and in vivo digestion of persimmon and derived products: A review" clearly establishes the nutritional and economic importance of the persimmon fruit and reports on the scientific findings obtained for the fruit and its derivatives, particularly those related to gastrointestinal digestion and potential health effect. The manuscript is clear and has a logical sequence in its structure.

The tables only need to include the definitions of the abbreviations used as table notes. In addition, more specific information on the bioactive relationship and the microbiota should be incorporated.
Some minimal suggestions for improving the manuscript are presented below:

For table 1 define at the bottom of the table EP, NEP, HAD and FD

For Table 2, define TC, HDL, LDL, and TG at the bottom of the table.

In line 234-236 The authors could provide more information on the modification of bacterial genera due to the presence of polyphenols and write this information in the paragraph.

Author Response

The manuscript entitled "In vitro and in vivo digestion of persimmon and derived products: A review" clearly establishes the nutritional and economic importance of the persimmon fruit and reports on the scientific findings obtained for the fruit and its derivatives, particularly those related to gastrointestinal digestion and potential health effect. The manuscript is clear and has a logical sequence in its structure.

The tables only need to include the definitions of the abbreviations used as table notes. In addition, more specific information on the bioactive relationship and the microbiota should be incorporated.
Some minimal suggestions for improving the manuscript are presented below:

For table 1 define at the bottom of the table EP, NEP, HAD and FD

The definition of EP, NEP, HAD and FD has been added at the bottom of the Table 1 in the new manuscript version. (Line 83)

For Table 2, define TC, HDL, LDL, and TG at the bottom of the table.

The definition of TC, HDL, LDL and TG has been added at the bottom of Table 2 in the new manuscript version. (Line 337)

In line 234-236 The authors could provide more information on the modification of bacterial genera due to the presence of polyphenols and write this information in the paragraph.

New Information has been added. (Lines 233-243)

This manuscript is a resubmission of an earlier submission. The following is a list of the peer review reports and author responses from that submission.

Round 1

Reviewer 1 Report

This review was a very interesting read on recent findings about persimmons and persimmon-derived products, particularly focusing on their expected physiological points. This article seems to have enough content as a review. The parts that are considered to need revisions are written below, so please refer to them and correct them.

Line 65.  That condition the subsequent use….

Line 88.  TNO, Research2, and INRA3    unclear

   To make possible easy and clear understanding of the content of experiment, please prepare a flow chart, in which showed the times added substrates (food) and enzymes to the system (reaction mixture), the time (point) changed pH values of the reaction mixture, and the holding time at each step, along with the names of organs corresponding to each step.  So, it is recommended to prepare a chart as a figure.  Please write with full name of TNO and INRA3.  

Line 246~

  Please explain easily the analytical method, which used in the literatures for determination of EP and NEP contents. Is the difference in the polyphenol contents between before and after of digestion of persimmon (sample) define to NEP?  Definition of difference between EP and NEP is considered to be necessary to help understand this section .

Author Response

This review was a very interesting read on recent findings about persimmons and persimmon-derived products, particularly focusing on their expected physiological points. This article seems to have enough content as a review. The parts that are considered to need revisions are written below, so please refer to them and correct them.

Line 65.  That condition the subsequent use….

The sentence has been modified (Line 65)

Line 88.  TNO, Research2, and INRA3    unclear

   To make possible easy and clear understanding of the content of experiment, please prepare a flow chart, in which showed the times added substrates (food) and enzymes to the system (reaction mixture), the time (point) changed pH values of the reaction mixture, and the holding time at each step, along with the names of organs corresponding to each step.  So, it is recommended to prepare a chart as a figure.  Please write with full name of TNO and INRA3.  

The revised manuscript has been modified (lines 187-189). The flow chart has also been added (Figure 1).

Line 246~

  Please explain easily the analytical method, which used in the literatures for determination of EP and NEP contents. Is the difference in the polyphenol contents between before and after of digestion of persimmon (sample) define to NEP?  Definition of difference between EP and NEP is considered to be necessary to help understand this section.

An explanation has been added in the revised manuscript (Lines 257-261)

Reviewer 2 Report

This review deals with some generic aspects about some methodologies but it is not clear about what authors need to expressed in my personal opinion needs to be reviewed again.

Authors took a specific fruit to talk about general points and it is not necessary of impact.

I consider that the document needs to be rewritten and focus better in the aim of that authors expressed.

Author Response

This review deals with some generic aspects about some methodologies but it is not clear about what authors need to expressed in my personal opinion needs to be reviewed again.

This review aims to compile the current information on in vitro and in vivo digestion of persimmon and derived products, focusing on the main bioactive and nutritional compounds. For this purpose, some previous generic aspects and methodologies have been explained to let the readers know about the main characteristics of the fruit itself and the in vitroand in vivo digestion methods most used in the literature. Moreover, we consider that a prior explanation of the bioactive and nutritional compounds in persimmon is necessary. This can help the readers to understand the different determinations in the studies explained in the review.

Authors took a specific fruit to talk about general points and it is not necessary of impact.

We took this specific fruit because of its nutritional aspects, such as its high amount in phenolic compounds (especially tannins) with antioxidant activity. Moreover, this fruit is very popular in Mediterranean and Asian countries, and it is highly demanded in the market.

I consider that the document needs to be rewritten and focus better in the aim of that authors expressed.

In the new version of the manuscript, we have rewritten and focus better the aim of the review (lines 20-22, 73-75)

Round 2

Reviewer 2 Report

As I mention before, this manuscript is not clear about what authors are looking for, if it goes about persimmons it looks more as a review of the different methodologies for food digestion and fermentation, in my opinion it is not suitable to be published, or they change completely the focus or they write a review about methodologies for food digestion systems. But I can not see clearly the relevance of this document.